# Ultrasonic Welding of Acrylonitrile–Butadiene–Styrene Thermoplastics without Energy Directors

**DOI:** 10.3390/ma17153638

**Published:** 2024-07-23

**Authors:** Qian Zhi, Yongbing Li, Xinrong Tan, Yuhang Hu, Yunwu Ma

**Affiliations:** 1Shanghai Key Laboratory of Digital Manufacture for Thin-Walled Structure, Shanghai Jiao Tong University, Shanghai 200240, China; zhiqianhnust@163.com (Q.Z.); myw3337@sjtu.edu.cn (Y.M.); 2Hunan Engineering Research Center of Forming Technology and Damage Resistance Evaluation for High Efficiency Light Alloy Components, Hunan University of Science and Technology, Xiangtan 411201, China; tanxinrong0@163.com (X.T.); 18867379862@163.com (Y.H.)

**Keywords:** ultrasonic welding, ABS, weld formation, simulation, horn amplitude

## Abstract

Ultrasonic welding (USW) of thermoplastics plays a significant role in the automobile industry. In this study, the effect of the welding time on the joint strength of ultrasonically welded acrylonitrile–butadiene–styrene (ABS) and the weld formation mechanism were investigated. The results showed that the peak load firstly increased to a maximum value of 3.4 kN and then dropped with further extension of the welding time, whereas the weld area increased continuously until reaching a plateau. The optimal welding variables for the USW of ABS were a welding time of 1.3 s with a welding pressure of 0.13 MPa. Interfacial failure and workpiece breakage were the main failure modes of the joints. The application of real-time horn displacement into a finite element model could improve the simulation accuracy of weld formation. The simulated results were close to the experimental results, and the welding process of the USW of ABS made with a 1.7 s welding time can be divided into five phases based on the amplitude and horn displacement change: weld initiation (Phase I), horn retraction (Phase II), melt-and-flow equilibrium (Phase III), horn indentation and squeeze out (Phase IV) and weld solidification (Phase V). Obvious pores emerged during Phase IV, owing to the thermal decomposition of the ABS. This study yielded a fundamental understanding of the USW of ABS and provides a theoretical basis and technological support for further application and promotion of other ultrasonically welded thermoplastic composites.

## 1. Introduction

Lightweight materials have become essential in the automotive industry with the governmental regulations on energy savings and CO_2_ emission reduction. The application of lightweight materials, such as thermoplastic composites is considered an effective strategy in structural and manufacturing applications [1]. Acrylonitrile–butadiene–styrene (ABS) is regarded well for use in automobiles due to its advantageous combination of being lightweight, having a high specific strength and good processability [2]. In this context, an effective technique for joining ABS is imperatively important. Mechanical fastening, adhesive bonding, and welding are alternative methods to realize the permanent bonding of thermoplastic materials. The former two techniques have drawbacks of additional weight, stress concentration, complex pre/post-processing, etc., which hinder their further application in automobile manufacturing [3]. Ultrasonic welding (USW), a type of welding, has been demonstrated to be suitable for the production of polymeric joints with solid mechanical strength, but energy directors with various geometries, including triangular, semi-circular, or rectangular, are usually presented on the surfaces of adherends to concentrate the welding energy [4]. However, the existence of an energy director (ED) would easily bring defects, such as incomplete fusion or cracks [5]. Therefore, the ultrasonic welding of thermoplastic materials without EDs has become a research hotspot.

To date, the researches on the USW of thermoplastics without ED mainly focus on the optimization of the welding variables and understanding the heating generation mechanism. Gao [6] et al. investigated the effects of welding parameters on joint strength, microstructure and weld appearance, and concluded that an ultrasonically welded carbon fiber-reinforced polyamide 66 (CF/PA 66) joint without ED can obtain a tensile strength of 5.2 kN with acceptable cosmetic quality and fine compact weld microstructure. Several researchers were devoted to enhancing the mechanical properties of the joint by heat treatment. Prior to the USW of a CF/PA composite, preheating [7] and annealing treatments [8] were applied and the tensile strength increased by approximately 40% compared to that of a normal joint. This phenomenon was correlated with the viscoelastic behavior and crystallinity of the thermoplastic, in which the proper heat pretreatment temperatures (95–125 °C for CF/PA 66 and 180 °C for CF/PA 6) concentrated the welding heat at the contact surface and decreased the dissipation within the workpieces. Accordingly, a proper weld without an obvious porous area was formed. Others also applied mechanical grinding [9], a blank holder [10,11] and a double pulse [12] to improve the welding conditions for the enhancement of tensile strength. These pretreatments aimed to ameliorate the contact behavior at the workpiece/workpiece interface prior to welding, and a robust tensile strength with small data variance was simultaneously obtained.

Finite element analysis (FEM) is a helpful tool in analyzing the weld formation in the USW of polymers, which can provide a comprehensive insight into the dynamic deformation behavior of the workpieces, and the strain/stress distribution at the overlapped region is also possible to predict. Currently, most FEM-related works are concentrated on revealing the heat generation mechanism during the USW process. Tutunjian [13] declared the frictional heat generated in the early stages of the ultrasonic spot welding of 5-HS-woven carbon fabric-reinforced thermoplastic through explicit mechanical 3D FEM analysis. The composite laminates were defined using a 3D continuum shell (only displacement degrees of freedom and no rotation) with eight nodes and reduced integration with the hourglass effect. Friction heating reduces the stiffness of the interface layers inside the weld center, and the applied cyclic strain focuses on the softer interfacial layers and induces a much higher viscoelastic heat. Consequently, the heating process is accelerated and restricted to the weld center with the combination of friction and viscoelastic heat. Li [14] proposed an integrated process–performance model to predict the failure load of the CFRP joint and weld formation information. It was verified that the heat generation mechanisms in the USW of thermoplastics came from friction and viscoelastic dissipation.

In practical USW, the ultrasonic oscillations change quickly, and transient vibrations are difficult to simulate. Currently, several researchers have applied a static pressure and constant horn displacement [14] on the tip of the horn or considered horn displacement changes step by step [15]. However, the experimental and simulated results have shown large differences. The horn vibration during USW was complicated not only because of its high frequency but also due to the resonance effect [16,17]. To date, there is scant research simulating the weld formation in the USW of thermoplastics by applying real-time vibrations in an FEM model. Therefore, there is an urgent need to understand weld initiation and growth by FEM analysis with real-time oscillations.

In this context, the actual vibrations during the USW of ABS were collected by a high-frequency sensor (400 kHz). The real-time oscillations were integrated into the FEM model to investigate the heat generation and weld formation. The effect of the welding time on joint performance and weld area evolution was assessed. The experimental results were also compared with the simulated results.

## 2. Materials and Methods

### 2.1. Laminate

ABS laminate with dimensions of 100 mm in length, 30 mm in width and 2 mm in thickness was purchased from Changzhou Xinhejiu Composite Materials Technology Co., Ltd. (Changzhou Xinhejiu Composite Materials Technology, Changzhou, China). The properties of ABS provided by the supplier (25 °C) were listed in Table 1.

### 2.2. Ultrasonic Welding

All the ABS laminates were joined on a KZH-2026 welder (Weihai Kaizheng Ultrasonic Technologies Co., Ltd., Weihai, China) with a nominal frequency of 20 kHz, and a nominal amplitude of 25 μm. The welder had three working modes of time-, energy- and displacement-control, and time-control was selected in this experiment. Prior to USW, the delay time, welding time and holding time were preset. The delay time (from the initial welding process to ultrasonic oscillations began) and holding time (ultrasonic oscillations stopped to the welding horn retracted back) were set as 2 s and 3 s on the basis of preliminary experiments, respectively [18]. Three high-frequency sensors—displacement, pressure and power sensors—at 400 kHz were installed on top of the pneumatic motion axis in the welder to collect real-time USW process information. Then, a data acquisition system (>500 kHz) connected the end of the sensor with a computer as shown in Figure 1a. A single lap joint with an overlap distance of 25 mm was applied.

### 2.3. Characterization

Tensile testing of the joint was conducted on an MTS E45.105 tester (MTS, Prairie, MN, USA) with a stroke rate of 2 mm/min based on ASTM standards [19]. Two filler plates were attached to the ends of the workpieces to accommodate the offset as shown in Figure 1b. Five sets of joints were welded with identical welding variables and the average joint strength (peak load) was obtained. The microstructure of the joint was examined with scanning electrical (SEM, SU3500 Hitachi, Tokyo, Japan). Prior to the examination, the sample was sputter-coated with gold to increase conductivity.

### 2.4. Rheological Experiment

The rheological experiment was carried out to measure the viscoelastic properties of ABS by using a shear-strain-controlled rotational rheometer (ARES-G2, Waters, Milford, MA, USA). The ABS specimen, in a circular shape of coin size, was subjected to a frequency sweep in the temperature range from room temperature to 180 °C with an ascending temperature rate of 2 °C/min to study its temperature/frequency-related performance. The storage modulus and loss modulus were obtained as a function of frequency, with a shear strain of 1% by applying the angular frequency in the range between 10^−1^ and 10^2^ rad/s.

## 3. Modelling

### 3.1. Material Property

ABS polymer is a typical viscoelastic material, and the Maxwell or Voigt–Kelvin model is usually utilized to describe the elastic and viscous properties. The Maxwell model, consisting of a spring (representing the elastic part) and a dashpot (denoting the viscous part), is selected in this simulation. Ten Maxwell units in parallel are used as also reported in other literature [19].

The constitutive relation of viscoelastic polymer is needed to understand the stress and deformation of ABS. Combing the Maxwell model, the constitutive behavior of ABS can be defined using [20]:(1)$$\sigma \left(t\right)=\varepsilon_{0}e\left(t\right)+\int_{0}^{t}e\left(t-\lambda \right)\frac{d\varepsilon \left(\lambda \right)}{d\lambda }d\lambda$$
where *σ(t)* is stress, ε_0_ is the initial value of strain, *t* and *λ* are the current and past time, respectively. *e(t)* is the relaxation modulus. The experimental frequency-related data are input into the model to define the viscoelastic property of the material. To connect the experimental test of storage and loss modulus with the constitutive model of ABS, the Maxwell series is converted to the frequency domain from the time domain using the Fourier transform and can be expressed as follows [21]:(2)$$E^{\prime }\left(\omega \right)=G_{0}\left[1-\sum_{i}^{N}g_{i}\right]+G_{0}\sum_{i=1}^{N}\frac{e_{i}\lambda_{i}^{2}\omega^{2}}{1+\lambda_{i}^{2}\omega^{2}}$$
(3)$$E\prime \prime \left(\omega \right)=G_{0}\sum_{i=1}^{N}\frac{e_{i}\lambda_{i}^{2}\omega^{2}}{1+\lambda_{i}^{2}\omega^{2}}$$
(4)$$G_{0}=\frac{e_{0}}{2\left(1+\mu_{0}\right)}$$
where *ω* is the angular frequency, *G*_0_ is the transient shear modulus, *μ* is the transient Poisson ratio, and *N* is the number of Maxwell series. The relaxation spectra of ABS in the range from 10^−1^ to 10^2^ rad/s are shown in Figure 2.

In defining the viscoelastic property of the material, which varies with frequency in Abaqus, the storage/loss moduli-related real and imaginary parts of $\omega ℜ\left(g^{*}\right)$ and $\omega ℑ\left(g^{*}\right)$, and the volume modulus-related $\omega ℜ\left(k^{*}\right)$ and $\omega ℑ\left(k^{*}\right)$ are needed. Since the ABS composite is a viscoelastic polymer, the imaginary parts are neglected [22].
(5)$$\omega ℜ\left(g^{*}\right)=\frac{E\prime \prime }{G_{\infty }}$$
(6)$$\omega ℑ\left(g^{*}\right)=1-\frac{E^{\prime }}{G_{\infty }}$$
(7)$$G_{\infty }=G_{0}[1\;-\sum_{i}^{N}e_{i}]$$
where *G_∞_* is the long-term shear, with 789.09 MPa for ABS material.

The temperature dependence of the materials can be considered with the WLF model [21]:(8)$$-log\alpha_{T}=\frac{C_{1}\left(T-T_{0}\right)}{C_{2}+\left(T-T_{0}\right)}$$
where *α_T_* is the horizontal shift factor, *T* is the temperature, *T*_0_ is the reference temperature and chosen as 180 °C to generate the master curve, and *C*_1_ and *C*_2_ are the fitting parameters, with *C*_1_ = 5.8 and *C*_2_ = 120.8 K in this study.

After defining the viscoelastic property of the material, the heat-transfer-related property of heat capacity, which changes with temperature, is considered. The specific heat (*c*) is expressed as follows [23]:(9)$$c=\frac{1}{m}\times \frac{dQ/dt}{dT/dt}$$
where *m* is the mass of the tested material, dQ/dt is the heat flux and dT/dt is the heating rate during the DSC test. The specific heat curve of ABS varies with temperature as shown in Figure 3 (blue line). The thermal gravity curve is also included in Figure 3, where melting and decomposition of ABS occur at 220 °C and 265 °C (intersections of the green dotted line and black solid line), respectively. Therefore, careful selection of the welding time is crucial in the USW of ABS.

### 3.2. Finite Element Modelling

The commercial Abaqus software (version 6.22) is implemented for the finite element modelling of the ultrasonically welded ABS process, and this software enables one to perform geometric modelling, material property definitions, meshing, and visualization. Then, the temperature evolution and stress distribution can be simulated.

In this study, a three-dimensional finite element model, consisting of a 7075 aluminum horn, aluminum anvil and two pieces of 2 mm-thick ABS laminates, is built in Abaqus as shown in Figure 4. The C3D8T hexagonal solid element is utilized to divide the mesh. The total number of meshed grids of the model is 5724, consisting of 9142 nodes. Considering the calculation accuracy and time taken for the analysis, the mesh size for the overlapped region between upper and lower workpieces is 1 while that for the rest of the regions is set as 5.

Thermal–mechanical coupling analysis is applied to simulate the temperature field in the USW of ABS. The fixture is fixed in the X-/Y-/Z-direction. The workpieces are fixed in the X- and Z-direction, with a small gap of 0.1 mm between the workpiece and fixture set as a boundary condition, and are illustrated in Figure 4. To improve simulation accuracy, real-time vibration (unveiling the amplitude change) during the USW process, which is recorded using a high-frequency sensor and data collector software installed in the welder, is introduced into the model and applied on the tip of the horn.

The initial temperature of the model is set as 20 °C, and the governing equation of heat conduction is expressed as [24]:(10)$$\rho c\frac{\partial T}{\partial t}=\frac{\partial }{\partial x}\left(\lambda \frac{\partial T}{\partial x}\right)+\frac{\partial }{\partial y}\left(\lambda \frac{\partial T}{\partial y}\right)+\frac{\partial }{\partial z}\left(\lambda \frac{\partial T}{\partial z}\right)+Q$$
where *ρ* is the material density, *λ* is the thermal conductivity of ABS, and *Q* is the welding energy. As for heat convection, the convection boundary conditions between the horn and upper workpiece, lower workpiece and the fixture are set as 90 W/(m^2^ K) since the horn and fixture are both made of 7075 aluminum alloy. The contacting heat conduction coefficient between upper and lower workpieces (*R*) is defined using [25]:(11)$$R=\frac{KA}{L}$$
where *K* is the thermal conductivity of ABS, with a value of 0.226 W/m/K, *L* is the sheet thickness of 2 mm, and *A* is the sectional area where the ABS sample is perpendicular to the conduction direction. The contact area in this model is the overlapped region, with a sectional area of 900 mm^2^. The calculated contacting thermal conductivity coefficient *R* is 0.1 W·m^−2^·K^−1^. Heat conduction between the components and ambient atmosphere is not considered owing to the short welding time (less than 2 s).

Surface-to-surface contact is utilized to define component contact in the model. Hard contact is applied when it is normal to the workpiece where a separation is allowed during the USW process. A penalty function is used to define the friction between two parts. Friction coefficients at the horn to upper workpiece and lower sheet to fixture interfaces are set as 0.1 [7,26,27]. At the workpieces contacting surface, the friction coefficient is 0.3 when the temperature at the contacting interface is below the melting point of ABS (220 °C) while it is defined as 0.1 with a further increase in temperature.

## 4. Results and Discussion

### 4.1. The Joint Strength of Ultrasonically Welded ABS

Preliminary experiments show the welding pressure of 0.13 MPa is more suitable for the USW of ABS. Herein, the effect of the welding time on peak load and weld area of the joint welded with a welding pressure of 0.13 MPa is evaluated as shown in Figure 5. With the increase in the welding time, the peak load of the joint increases to the maximum value of 3.4 kN and then decreases, while the weld area (measured by the IPP 6.0 software) expands gradually to a plateau. A peak load of 0.37 kN is obtained for the joint with a 0.1 s welding time, and the joint strength and weld area increase dramatically in the first 0.9 s then increase moderately to their peak values. Combining the peak load and weld area, the optimal welding time is selected as 1.3 s, where it can reach the maximum peak load and weld area. Interestingly, the data variance in joint strength becomes larger when the welding time is extended to above 1.3 s. This characteristic is likely attributed to the thermal decomposition of ABS, where the position and distribution of the resultant porous region affect the joint strength severely [6,28].

### 4.2. Microstructure of the Joint

The fractured surfaces of the joints after tensile testing are observed and two main fracture modes of interfacial failure and workpiece breakage are presented as shown in Figure 6. The joint usually fractures at the nugget for underweld joints (welding time shorter than 1.1 s) where the insufficient weld area cannot bear enough tensile force. Prolonging the welding time to above the optimal value, the joint is likely to break at the workpiece (mostly at the upper workpiece), which is closely related to the material property [28] and will be elaborated on later.

Careful examination of the fractured morphology of the joint shown in Figure 7 shows that the weld area expands with the welding time and there is a loose microstructure with pores on the surface when it exceeds the optimal time. The effective bonding region is localized in an approximately circular area (directly under the welding horn) and the bonding area can be categorized into two types: compact normal weld area and loose porous area. The microstructure in the normal weld area is dense, while numerous deconsolidation voids appear in the central area of the weld region and are labelled as porous regions in Figure 7. Cross-sectional morphologies of representative joints welded in 1.1 s, 1.3 s, 1.5 s and 1.7 s are observed to analyze the weld nugget. Referring to Figure 8, there is no obvious boundary between the weld zone and matrix when the welding time is less than 1.1 s. The distinction of the weld region is mostly based on the direction of the fracture texture, where the stress conditions for upper and lower workpieces during the tensile test are different. Increasing the welding time to 1.3 s, some pores randomly distribute in the weld zone. The pore increases in density and quantity. The thickness of the porous layer increases with increased welding time, which is harmful to the joint [7,29]. For joints made in 1.7 s, the scale of the porous region enlarges remarkably, and it exhibits a mouth-like shape, where the thickness of the lateral side is small and the central region is relatively large. This behavior is intimately associated with heat generation and melt flow during USW [12,18,29], where thermal decomposition of the polymer releases gases. The pressure in the gases is large, and large pores extend to the faying interface and squeeze the molten materials out. The decomposed material flows bilaterally along the weld area and forms a mouth-like shape (joints with a 1.7 s welding time).

A small number of pores in the central region of the weld area slightly influence joint performance, as reported previously [6,28]. Increased pores in the weld zone separate the polymer matrix and weaken its ability to bear loading. Thus, the maximum peak load occurs at a welding time of 1.3 s and drops afterwards.

### 4.3. Simulation of the USW Process

#### 4.3.1. Heat Generation

To fully understand the weld formation of ultrasonically welded ABS, the joints made with 0.13 MPa and 1.7 s are simulated using Abaqus 6.22 software, and the side views of the simulated weld with various welding times are shown in Figure 9. The melting temperature of the matrix is assigned as T_m_ while the thermal decomposition point is denoted as T_d_. The regions in red or grey color represent the polymer melts or decompose under ultrasonic heat. The highest temperature at the contact interface is in the middle of the overlapped region. Surprisingly, the polymer melts with only a 0.1 s welding time, which is much faster than that of polyamide composites [6,18]. This characteristic verifies that the amorphous polymer is easier to weld than the a polymer with a semi-crystal structure [29,30,31]. With increased welding time, the red melted region expands gradually into the X/Y/Z plane. The grey decomposed region emerges at joints made in 1.1 s and displays an analogous expansion trend with the melted region.

The theory that the heat generation in the USW of thermoplastics generally contains frictional heat and viscoelastic dissipation has been well accepted by scholars [16,19,32]. The frictional heat and viscoelastic dissipation in the USW of ABS are derived using “process output” in Abaqus, as presented in Figure 10. The varying tendencies of frictional and viscoelastic dissipations are contrary, which is consistent with the opinion that frictional heat dominates at the early stage of the USW while viscoelastic heating dominates in the following stages [9,19,24].

At the initial stages of USW, the upper and lower workpieces retain their stiffness and toughness, along with the rough surface of the adherends. Before the steady melt forms, the perpendicular vibrations transfer to horizontal deformation at the interface of workpieces. Horizontal deformation adds to slippage and friction at the interface. Friction generates heat on the rough surfaces of workpieces. Once the weld nugget forms, frictional heat greatly decreases. When the temperature of the workpiece rises up to above the glass transition temperature under friction heat, viscoelastic dissipation dominates the ultrasonic welding process, as shown in Figure 10 (blue lines).

#### 4.3.2. The Weld Formation Mechanism

Real-time vibrations for joints with a 1.7 s welding time (the semi-transparent grey region) are also included in Figure 10. As seen from the enlarged view of horn displacement, the vibrations are complicated and change quickly during USW. Hence, simulating weld growth of ultrasonically welded ABS by applying real-time vibration into the model should primarily improve simulation accuracy. Horn displacement in Figure 10 exhibits typical characteristics in the USW of thermoplastics [33,34], and the weld area of the simulated area and measured areas shows similar varying trends in Figure 11, Figure 12, Figure 13 and Figure 14 as expected. Comprehensively considering the variation characteristics in amplitude, vibration, horn displacement and energy dissipation, the weld formation of ultrasonically welded 2 mm-thick ABS without energy can be divided into five phases.

Phase I (0~0.1 s): This phase lasts approximately 0.1 s and the fractured surface shows a clear weld initiation with random hot spots as depicted in Figure 11. At this stage, the melting of the matrix is mainly attributed to friction heating, which results from contact point slippage [18]. Since this phase is very short and most of the matrix remains un-melted, no significant horn displacement or indentation is observed. Thus, the amplitude in phase I is larger than the nominal amplitude owing to the resonance in the system [27].

Phase II (0.1~0.6 s): This stage is an unsteady phase. The amplitude changes irregularly and the small asperities at the contact surface melt gradually to form a favorably intimate contact condition for the following phase. Fiction and viscoelastic heat together dominate weld growth at this stage [18]. The heat reduces the stiffness of the interface layers inside the weld apex and the sinusoidal cyclic strain focuses on the softer interfacial area to expand the weld area, as shown in Figure 12. The simulated result is smaller than the measured one owing to the flow and expansion of the melt. A slight horn indentation is presented on the joint. A downward trend in displacement at Phase II is presented in Figure 10, which is mainly due to the thermal expansion of the ABS matrix under the accumulation of ultrasonic heat and causes horn retraction.

Phase III (0.6~1.3 s): The amplitude change is relatively stable in this phase and is characterized by a continuously increasing displacement. The melt rate and flow of the ABS matrix are in equilibrium. Then, weld growth enters into the paramount phase—steady melt flow—which is critical to the joint quality [19,35]. It is worth mentioning that there is a small step-like shape to the smoothed displacement. This phenomenon is correlated with melt flow behavior. It is seen in Figure 10 that viscoelastic heating plays a chief role in weld growth. The simulated weld growth at this stage is also smaller than the measured weld growth as explained in Phase II, while the squeeze out of molten ABS and horn indentation become significant when the welding time exceeds 1.3 s, as illustrated in Figure 13. Before this meltdown, a large amount of ABS melts within the upper plate and is about to be squeezed out. Hence, there is a slight change in the amplitude.

Phase IV (1.3~1.7 s): This phase is characterized by a combination of horn indentation on the upper workpiece and squeezed out of molten ABS. The amplitude at this stage is much more stable and the displacement of the horn increases linearly with the welding time (with a larger increase rate than that in Phase III). At this phase, more viscoelastic dissipation is consumed within the workpiece and leaves numerous voids in the weld region and deep indentation on the joint surface (Figure 14). To thoroughly understand the origin of the pores, Fourier transform infrared spectroscopy (FTIR) tests are conducted in the ABS matrix, weld area and porous region. Referring to Figure 15, the bending vibration of C-H, deformation of C-H for hydrogen atoms and out-of-plane C-H bending in ABS polymer are in the range of 700–1038 cm^−1^. Stretching vibration peaks of the benzene ring, C≡N and C=C are present at 1450–1600 cm^−1^, 2237 cm^−1^ and 1630 cm^−1^, respectively. The aromatic and aliphatic C-H are detected at 3200–2800 cm^−1^ [36,37]. It is clear that the peak positions are similar but the absorption intensity differs significantly. The peak intensity for the porous area is much lower, but similar for the weld area and the matrix, indicating the ABS material decomposes in the porous area [28,38]. Generally, a solid joint should have a dense microstructure, thus the occurrence of this phase is detrimental to the joint strength and should be avoided in actual production.

It has been verified that the voids result from the thermal decomposition of the ABS matrix. The decomposed ABS releases volatile products, such as HCN, CO, and NO_x_. The pressures in these voids are large and will be expanded to the welding interface, accompanied by the squeeze out of molten ABS. With the accumulation of viscoelastic dissipation, the upper workpiece (dissipated the majority of the heat) decomposes consequentially and some of the decomposed ABS flows bilaterally along the weld apex, while the rest of the residues are in the joint. As a result, the joint with a welding time of 1.7 s shows a microstructure with a mouth-like shape and numerous pores, as presented in Figure 8.

Phase V (>1.7 s): Ultrasonic vibration stops at this stage and horn displacement is slightly increased owing to the cold contraction of the weld. The weld solidifies under the welding pressure for 3 s (holding time).

Based on the aforementioned analysis, the characteristics of each phase during the USW of ABS are different. Weld initiations with randomly distributed hotspots are observed in Phase I. The unsteady and steady phases have a sequential pattern with a downward and increasing trend in horn displacement, respectively. Then, thermal decomposition of ABS occurs, with faster and increased horn displacement. When ultrasonic vibration is paused, the process enters Phase V with a slight increase in horn displacement.

## 5. Conclusions

The USW of ABS without ED was investigated in this study. The effects of the welding time on joint performance, weld area and the weld formation mechanism were analyzed systematically. The following main conclusions were drawn:(1)The peak load of ultrasonically welded ABS increased with the welding time (less than 1.3 s) and then decreased with a prolonged welding time. The maximum value of 3.4 kN was obtained with an optimal welding time of 1.3 s and 0.13 MPa.(2)On prolonging the welding time to 1.7 s, the weld areas of joints increased gradually to the maximum value and then reached a plateau. Two typical failure modes of interfacial failure and workpiece breakage appeared during tensile tests.(3)Integrating real-time horn displacement into the finite element model can improve simulation accuracy in the USW of ABS.(4)Weld formation of ultrasonically welded ABS without ED (welding time of 1.7 s) consisted of five distinct phases of weld initiation, horn retraction, melt and flow equilibrium, horn indentation and squeeze out, weld solidification based on the variation characteristics, horn displacement and energy dissipation during welding.(5)An obvious porous area emerged in the joint made with a welding time greater than 1.3 s, which was mainly ascribed to the thermal decomposition of ABS and was detrimental to the joint strength.

## Figures and Tables

**Figure 1 materials-17-03638-f001:**
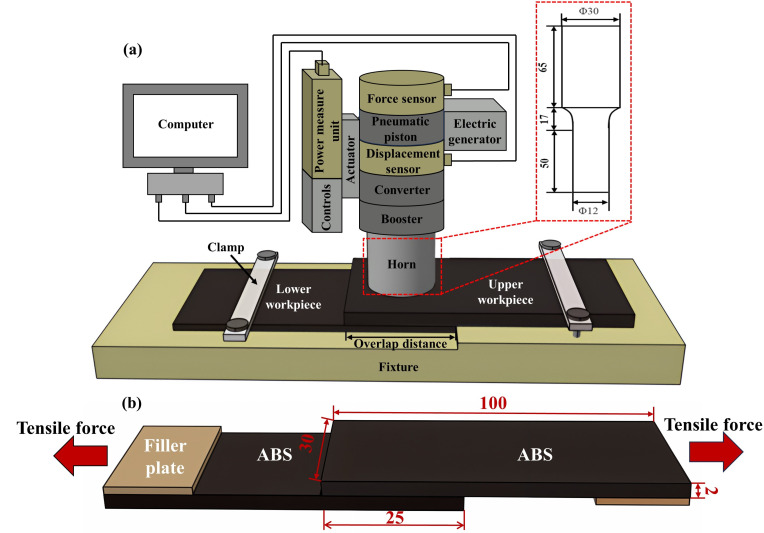
Schematic illustration of ultrasonically welded ABS without energy directors: (**a**) welding configuration; (**b**) single-lapped joint (dimensions in mm).

**Figure 2 materials-17-03638-f002:**
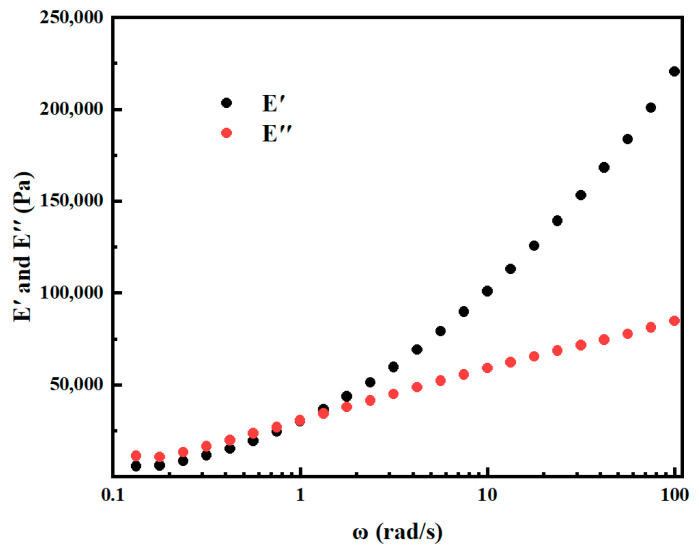
Discrete relaxation spectra of ABS in the frequency range of 0.1–100 Hz.

**Figure 3 materials-17-03638-f003:**
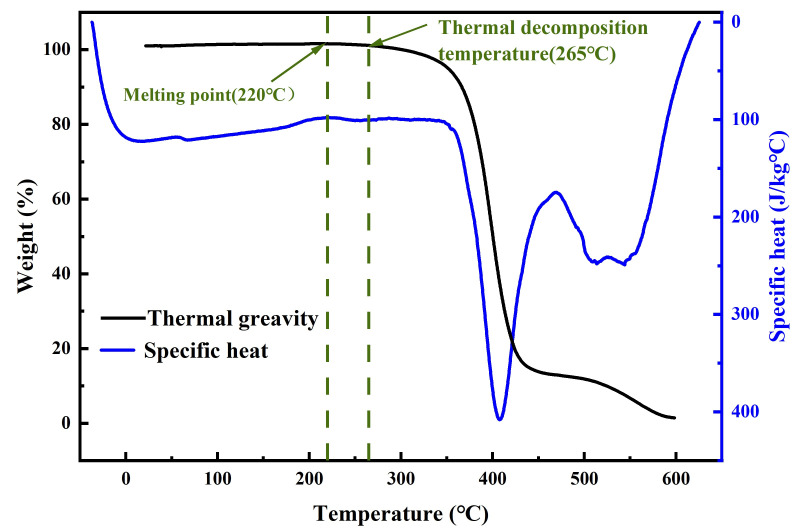
Heat-specific and weight loss curves of ABS.

**Figure 4 materials-17-03638-f004:**
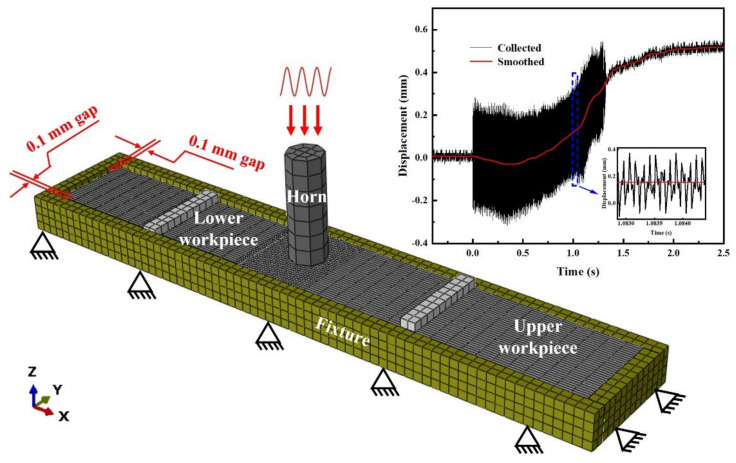
Finite element model (integrating transient vibrations which presented at top right of the figure into model) of ultrasonically welded ABS without energy directors.

**Figure 5 materials-17-03638-f005:**
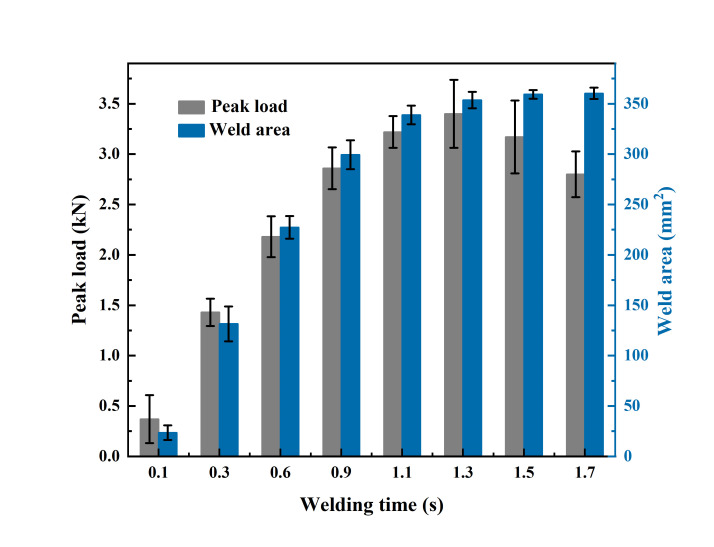
Effect of the welding time on peak load and weld area of ultrasonically welded ABS.

**Figure 6 materials-17-03638-f006:**
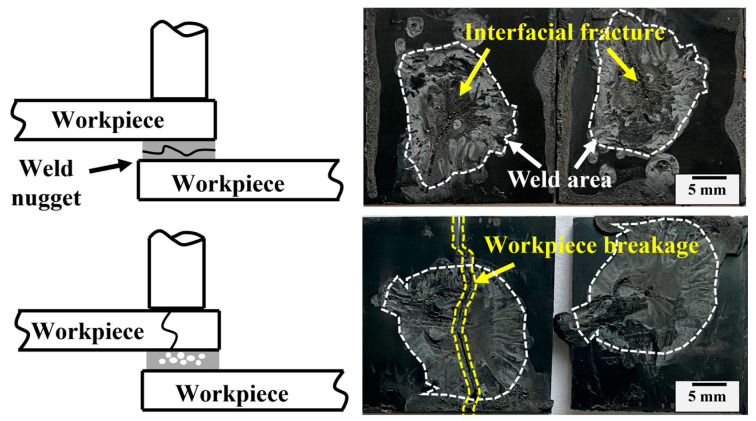
Fracture mode of ultrasonically welded ABS joint.

**Figure 7 materials-17-03638-f007:**
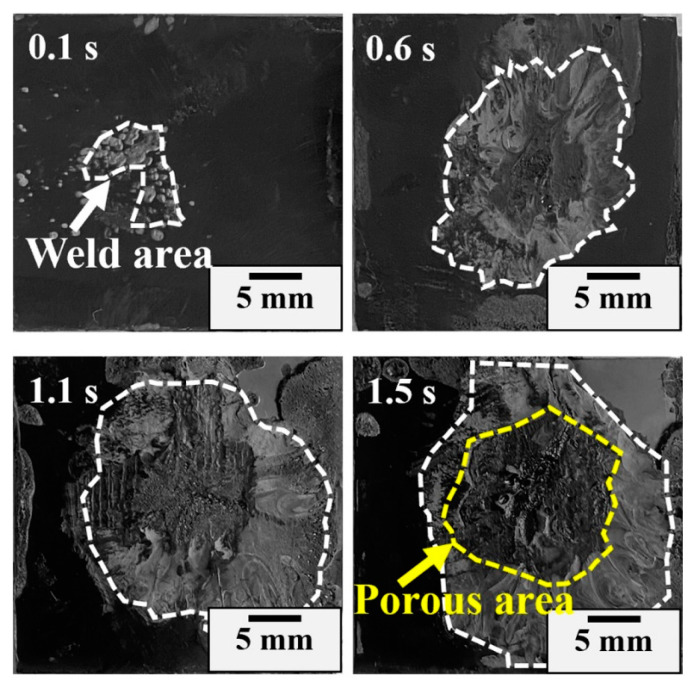
Relation between the welding time and weld area.

**Figure 8 materials-17-03638-f008:**
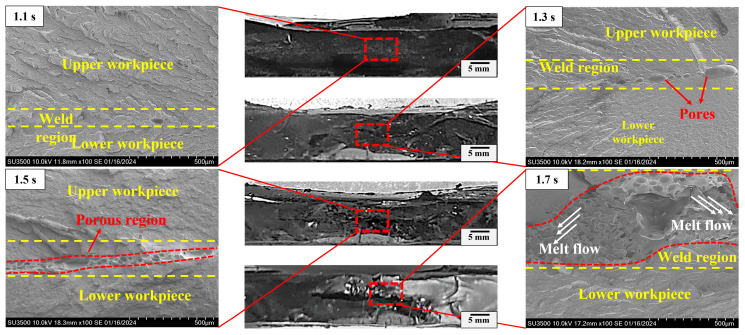
Cross-sectional morphology of representative joints.

**Figure 9 materials-17-03638-f009:**
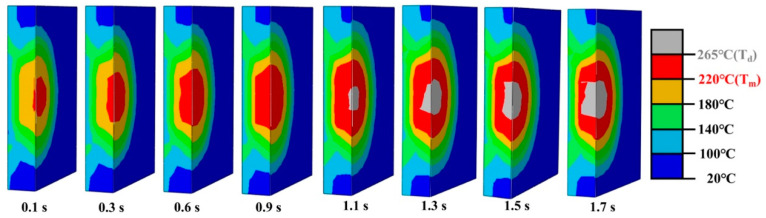
Simulation temperature distribution in the upper workpiece of the joint welded with different times.

**Figure 10 materials-17-03638-f010:**
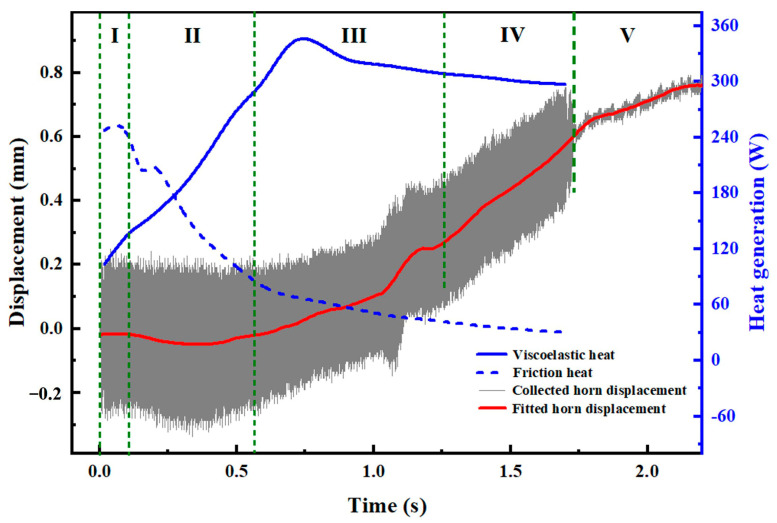
Heat generation and horn displacement curves in ultrasonically welded ABS in 1.7 s.

**Figure 11 materials-17-03638-f011:**
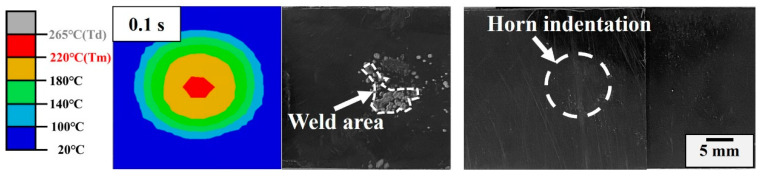
Simulation and experimental weld area; horn indentation of the welded joint in Phase I.

**Figure 12 materials-17-03638-f012:**
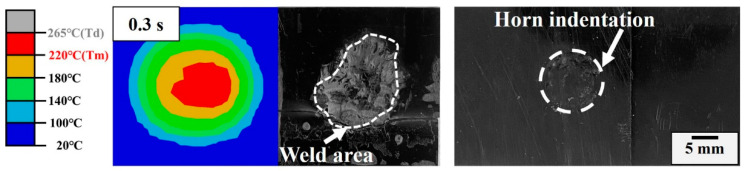
Simulation and experimental weld area; horn indentation of the welded joint in Phase II.

**Figure 13 materials-17-03638-f013:**
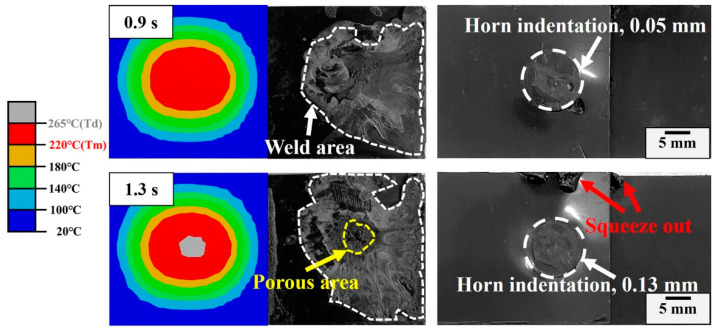
Simulation and experimental weld area; weld appearance of the joint in Phase III.

**Figure 14 materials-17-03638-f014:**
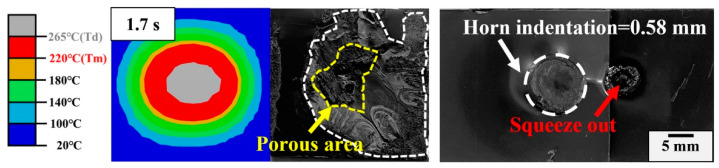
Simulation and experimental weld area; weld appearance of the joint in Phase IV.

**Figure 15 materials-17-03638-f015:**
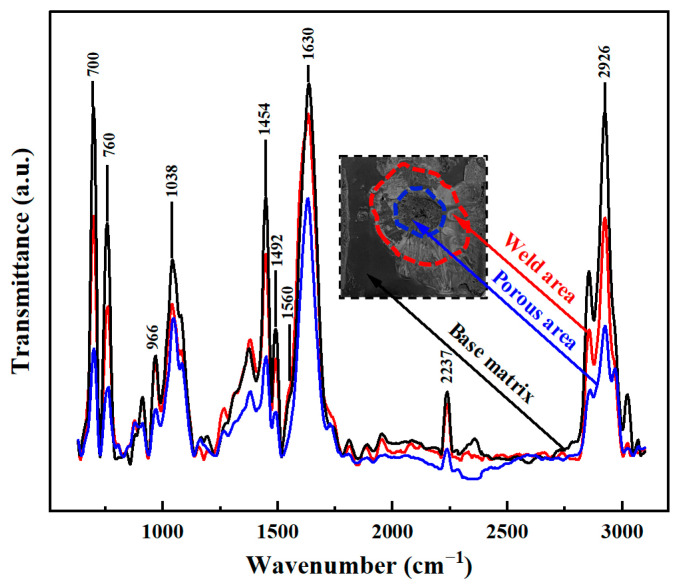
FTIR spectra of the joint within different regions.

**Table 1 materials-17-03638-t001:** Properties of ABS laminate.

Materials	Density/(kg · m^−3^)	Poisson’s Ratio	Elastic Modulus/(MPa)	Thermal Conductivity/(W · m^−1^K^−1^)
ABS	1100	0.394	2000	0.2256

## Data Availability

The original contributions presented in the study are included in the article, further inquiries can be directed to the corresponding author.

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
