# Peer review of "Ultrasonic Welding of Acrylonitrile–Butadiene–Styrene Thermoplastics without Energy Directors"

_materials, 2024, doi:10.3390/ma17153638_

Round 1

Reviewer 1 Report

Comments and Suggestions for Authors

The article is devoted to relevant and modern topics, but needs some improvement.

1. It would be nice to expand the introduction section a little. More clearly emphasize the novelty and importance of this area of ​​research. In the introduction, describe the finite element method in more detail and give examples of its application to analyze the mechanisms of formation and crystallization of a weld. Show the difficulties of conducting a full-scale experiment. Show the positive effect of replacing part of real experimental work with modeling.

2. It is necessary to add a section on Methods and materials. In this section it is also worth showing in more detail the methodology for conducting a rheological experiment. Indicate the limits of change in the initial data (data from Table 1).

3. In the Methodology section, show the version of the software used, briefly show the capabilities of this software.

4. When analyzing the work results, show mathematical models that allow you to calculate welding time based on the data presented in the figures. How accurately can curing time be predicted?

5. Carry out a more complete analysis of the resulting heat release curves.

6. The conclusions should present specific practical results obtained in the work and prospects for conducting similar studies.

7. Improve the design of the article. In particular, captions must be made in the same style.

Author Response

July 4, 2024

Materials

Editorial Office

Dear Aleksandra Krakowka,Thanks for the comments for our manuscript entitled “Ultrasonic welding of ABS thermoplastics without energy director”, which was submitted to Materials for publication. The comments were highly insightful and would greatly help to improve the quality of our manuscript. We have carefully reviewed and prepared the answers for each question, and attached are our response and revised manuscript.

We would appreciate your efforts in reviewing our manuscript and shall look forward to hearing from your final decision when it is made.

Yours sincerely,

Qian Zhi

Shanghai Jiao Tong University, Shanghai, 200240, China

We have revised the manuscript to address the reviewer’s questions. All revisions are highlighted using red background in the revised manuscript.

  1. Response to Reviewer # Reviewer #1:
  2. The article is devoted to relevant and modern topics, but needs some improvement. It would be nice to expand the introduction section a little. More clearly emphasize the novelty and importance of this area of research. In the introduction, describe the finite element method in more detail and give examples of its application to analyze the mechanisms of formation and crystallization of a weld. Show the difficulties of conducting a full-scale experiment. Show the positive effect of replacing part of real experimental work with modeling.

Response 1:The reviewer’s comments are well-taken. The Introduction section has been improved in the revised manuscript.

  1. It is necessary to add a section on Methods and materials. In this section it is also worth showing in more detail the methodology for conducting a rheological experiment. Indicate the limits of change in the initial data (data from Table 1).

Response 2:The reviewer’s comments are well-taken. A dependent section of rheological experiment has been described in Section 2.4 Rheological experiment. The data listed in Table 1 is given by the supplier which is measured at 25 °C.

  1. In the Methodology section show the version of the software used, briefly show the capabilities of this software.

Response 3:The reviewer’s comments are well-taken. The version of the Abaqus is 6.22 and its capabilities are implemented in the revised manuscript.

  1. When analyzing the work results, show mathematical models that allow you to calculate welding time based on the data presented in the figures. How accurately can curing time be predicted?

Response 4:The welding time in the experimental results (Figs. 4 and 10) is read from the collected curves. As shown in Fig. 10, the ultrasonic oscillations are very different. Once the amplitude fluctuates in a relatively large range, donating the beginning of the process and when the amplitude changes slightly from the equilibrium position, the vibration stops. Since the ultrasonic vibration duration is preset in the welder, then the vibration fluctuation duration is defined as the welding time. The welding time is mathematical calculated, e.g., if the preset welding time is 1 second, and there are 1000 fluctuations (from the start to end of the vibration), then 1300 fluctuations equal to 1.3 s welding time, and so on.

Based on the previous preliminary experiments, the joint strength increases with the holding time firstly and then barely changes under the identical welding parameters. Combining efficiency and joint strength, we define the optimal holding time of 3 s as the curing time of the ABS polymer after ultrasonic welding and the related description and reference have been added in the revised manuscript.

  1. Carry out a more complete analysis of the resulting heat release curves.

Response 5:The reviewer’s comments are well-taken. Analysis of the heat release curve in Fig. 10 has been added in the revised manuscript. (the paragraph right before Fig. 10).

  1. The conclusions should present specific practical results obtained in the work and prospects for conducting similar studies.

Response 6:The reviewer’s comments are well-taken. Conclusions section has been rewritten.

  1. Improve the design of the article. In particular, captions must be made in the same style.

Response 7:The reviewer’s comments are well-taken. Captions have been revised in the same style.

Reviewer 2 Report

Comments and Suggestions for Authors

The author presents a work on ultrasonic welding of thermoplastic ABS. The manuscript shows a numerical finite element model that aims to portray the experimental welding with the best parameters that the author found, namely the welding pressure of 0.13 MPa and different welding times.

The work seems well structured, although I have some doubts that I think the author should answer, namely how he measured the area in fig. 5.

Another question is how the author validates his numerical model; It seemed to me that the description is not very justified.

In the attached pdf I present other questions.

Comments on the Quality of English Language

Please, see the file

Author Response

July 4, 2024

Materials

Editorial Office

Dear Aleksandra Krakowka,Thanks for the comments for our manuscript entitled “Ultrasonic welding of ABS thermoplastics without energy director”, which was submitted to Materials for publication. The comments were highly insightful and would greatly help to improve the quality of our manuscript. We have carefully reviewed and prepared the answers for each question, and attached are our response and revised manuscript.

We would appreciate your efforts in reviewing our manuscript and shall look forward to hearing from your final decision when it is made.

Yours sincerely,

Qian Zhi

Shanghai Jiao Tong University, Shanghai, 200240, China

We have revised the manuscript to address the reviewer’s questions. All revisions are highlighted using red background in the revised manuscript.

  1. Response to Reviewer # Reviewer #2:
  2. The author presents a work on ultrasonic welding of thermoplastic ABS. The manuscript shows a numerical finite element model that aims to portray the experimental welding with the best parameters that the author found, namely the welding pressure of 0.13 MPa and different welding times. The work seems well structured, although I have some doubts that I think the author should answer, namely how he measured the area in fig. 5.

Response 1:The weld area in Fig. 5 is measured using Image Pro Plus software. It is powerful 2D and 3D image processing, enhancement, and analysis software with extensive measurement and customization features. With Image-Pro Plus, one can count and characterize objects using manual and automatic measurement tools. It has been described in the revised manuscript.

  1. Another question is how the author validates his numerical model; It seemed to me that the description is not very justified.

Response 2:The reviewer’s comments are well-taken. The Results and discussion part has been revised according to the reviewer’s suggestion. The numerical model is validated by comparing the size of the weld area and cross-section morphologies of the joint as seen in Figs. 8, 11, 12, 13 and 14.

  1. In the attached pdf I present other questions.

Response 3:The reviewer’s comments are well-taken. Each suggestion has been carefully taken and revised in the revised manuscript.

Reviewer 3 Report

Comments and Suggestions for Authors

The article entitled Ultrasonic welding of ABS thermoplastics without energy director contains research on lap joints of thermoplastic material for applications in the automotive industry. The considerations concerned the influence of welding time on the joints' strength properties and the joint area's size. The results may be attractive due to the increasing share of plastics in technical applications, but I suggest making minor corrections before publication.
1.    Please add the quantitative results to the abstract.
2.    Introduction - Please provide information about the full spectrum of tests carried out, including strength properties.
3.    Experimental procedures - Please complete the information regarding the geometric parameters of the samples used in strength tests, as the results shown in Fig. 5 are difficult to compare with literature data.
4.    Conclusions: Please enter observations regarding the application aspects of the tests performed and add the percentage impact of time change on the weld strength.

Author Response

July 4, 2024

Materials

Editorial Office

Dear Aleksandra Krakowka,Thanks for the comments for our manuscript entitled “Ultrasonic welding of ABS thermoplastics without energy director”, which was submitted to Materials for publication. The comments were highly insightful and would greatly help to improve the quality of our manuscript. We have carefully reviewed and prepared the answers for each question, and attached are our response and revised manuscript.

We would appreciate your efforts in reviewing our manuscript and shall look forward to hearing from your final decision when it is made.

Yours sincerely,

Qian Zhi

Shanghai Jiao Tong University, Shanghai, 200240, China

We have revised the manuscript to address the reviewer’s questions. All revisions are highlighted using red background in the revised manuscript.

  1. Response to Reviewer # Reviewer #3:
  2. The article entitled Ultrasonic welding of ABS thermoplastics without energy director contains research on lap joints of thermoplastic material for applications in the automotive industry. The considerations concerned the influence of welding time on the joints' strength properties and the joint area's size. The results may be attractive due to the increasing share of plastics in technical applications, but I suggest making minor corrections before publication.

Response 1:The reviewer’s comments are well-taken and the revisions have been made in the revised manuscript.

  1. Please add the quantitative results to the abstract.

Response 2:The reviewer’s comments are well-taken. The abstract has been revised.

  1. Introduction - Please provide information about the full spectrum of tests carried out, including strength properties.

Response 3:The reviewer’s comments are well-taken. The introduction section has been revised and enriched.

  1. Experimental procedures - Please complete the information regarding the geometric parameters of the samples used in strength tests, as the results shown in Fig. 5 are difficult to compare with literature data.

Response 4:The reviewer’s comments are well-taken. The geometric parameters of the joint have been added in new revised Fig. 1b.

  1. Conclusions: Please enter observations regarding the application aspects of the tests performed and add the percentage impact of time change on the weld strength.

Response 5:The reviewer’s comments are well-taken and the conclusion section has been rewritten.

Reviewer 4 Report

Comments and Suggestions for Authors

1.       The conclusions must be included in the abstract, or at least some of the conclusions. The only conclusion is: “the simulated results were close to the experimental ones…”

2.       The main objective described at the end of the introduction is not the same as what appears in the summary. It must be corrected and it must be clear to readers what the objective of this study is.

3.       Table 1 – Given by the supplier? Obtained by authors?

4.       Experimental tests: according which standards? How many tests were performed at each condition?

5.       Shouldn't the analysis be done in terms of tension? What physical meaning does analysis have in terms of peak load?

6.       6. Were samples inspected before testing? Were there any pores or other defects?

7.       Line 238 – How can the authors justify this conclusion? Based on what?

8.       Has the temperature variation during the welding process not been evaluated? Why? wouldn't it be important for the discussion of the results and conclusions?

9.       Figure 9 - Are these conclusions only obtained based on numerical simulation? How can it be compared with the experimental ones? Are only the results of the numerical simulation enough to draw conclusions?

10.   Figure 15 is related to the FTIR result, but during the article no reference is made to FTIR.

11.   The results must be presented and discussed. Were the results expected or not? Why? Why not?

12.   Poor discussion with open literature. The conclusions may be questionable.

Author Response

July 4, 2024

Materials

Editorial Office

Dear Aleksandra Krakowka,Thanks for the comments for our manuscript entitled “Ultrasonic welding of ABS thermoplastics without energy director”, which was submitted to Materials for publication. The comments were highly insightful and would greatly help to improve the quality of our manuscript. We have carefully reviewed and prepared the answers for each question, and attached are our response and revised manuscript.

We would appreciate your efforts in reviewing our manuscript and shall look forward to hearing from your final decision when it is made.

Yours sincerely,

Qian Zhi

Shanghai Jiao Tong University, Shanghai, 200240, China

We have revised the manuscript to address the reviewer’s questions. All revisions are highlighted using red background in the revised manuscript.

  1. Response to Reviewer # Reviewer #4:
  2. The conclusions must be included in the abstract, or at least some of the conclusions. The only conclusion is: “the simulated results were close to the experimental ones…”

Response 1:The reviewer’s comments are well-taken and the Conclusions section have been rewritten in the revised manuscript.

  1. The main objective described at the end of the introduction is not the same as what appears in the summary. It must be corrected and it must be clear to readers what the objective of this study is.

Response 2:The reviewer’s comments are well-taken. The last paragraph in Introduction has been rewritten.

  1. Table 1 – Given by the supplier? Obtained by authors?

Response 3:The data in Table 1 is given by the supplier and has been explained in the revised manuscript.

  1. Experimental tests: according which standards? How many tests were performed at each condition?

Response 4:The reviewer’s comments are well-taken. Tensile test of the joint was conducted on an MTS E45.105 tester with a stroke rate of 2 mm/min based on ASTM D1002-2001. Five sets of joints (with the dimensions of 100 × 30 × 2 mm with an overlap of 30 mm as shown in Fig. 1b) were welded at each condition.

  1. Shouldn't the analysis be done in terms of tension? What physical meaning does analysis have in terms of peak load?

Response 5:The joint performance is evaluated through tensile test and peak load is selected in this study. It can be seen in this study (Fig. 6), there are two kinds of failure modes: interfacial and workpiece breakage. The joint strength for these two failure modes should use different areas, the former one using welding area while the latter applying the cross-section area. To unify and simplify the calculation, peak load is used to evaluate the joint performance. In our previous studies [1-4], peak load is always applied to evaluate the joint strength. To compare these results, peak load is selected in this study. Besides, there are some other scholars also use peak load to estimate the joint strength [5-8].

[1] Zhi Q, Tan XR, Liu ZX. Effect of moisture on the ultrasonic welding of carbon-fiber-reinforced polyamide 66 composite. Weld J 2017; 96(6): 185s–92s.

[2] Zhi Q, Tan XR, Liu ZX. Effects of preheat treatment on the ultrasonic welding of carbon-fiber-reinforced polyamide 66 composite. Weld J 2017; 96: 429s–38s.

[3] Zhi Q, Gao Y, Lu L, Liu Z, Wang P. Online inspection of weld quality in ultrasonic welding of carbon fiber/polyamide 66 without energy directors. Weld J 2018; 97: 65s–74s.

[4] Gao Y, Zhi Q, Lu L, Liu Z, Wang PC. Ultrasonic welding of carbon fiber reinforced nylon 66 composite without energy director. J Manuf Sci Eng 2018; 140(5): 051009.

[5] Wang K, Li Y, Banu M, Li J, Guo W, Khan H. Effect of interfacial preheating on welded joints during ultrasonic composite welding. J Mater Process Technol 2017; 246: 116–22.

[6] Choudhury MR, Debnath K. Analysis of tensile failure load of single-lap green composite specimen welded by high-frequency ultrasonic vibration. Mater Today Proc 2020; 28: 739–44, 2nd International Conference on Recent Advances in Materials Investigation on Ultrasonic Welding Attributes of Novel Carbon/Elium Composites Manufacturing Technologies.

[7] S. James, C. Dang. Investigation of shear failure load in ultrasonic additive manufacturing of 3D CFRP/Ti structures  J Manuf Process, 56 (2020), pp. 1317-1321

[8] Yang Y, Li Y, Liu Z, Li Y, Ao S, Luo Z. Ultrasonic welding of short carbon fiber reinforced PEEK with spherical surface anvils Compos Part B Eng, 231 (2022), Article 109599.

  1. Were samples inspected before testing? Were there any pores or other defects?

Response 6:Yes, each sample before welding is visually examined. The ABS samples are injection molded and have small variance in dimensions and planeness, thus the examination before welding is a must. By visual inspection, there are no visible pores or other defects. We also examined the samples using SEM technique randomly, and the microstructures are dense and compact.

  1. Line 238 – How can the authors justify this conclusion? Based on what?

Response 7:The melt flow behaviors with longer welding time have been discussed in our previous studies [9, 10, 11]. When extensive welding time is applied in ultrasonic welding of polymers (exceeds the optimal welding time), thermal decomposition of the polymer would occur and the decomposed composite releases some gases. These gases (pores) could not have been instantly expelled out. The pressure in these pores was large, and the large pores extended to the faying which squeezed the molten materials out. The decomposed material flows bilaterally along the weld area and forms a mouth-like shape. The related references have been cited and simple explanation of this characteristic has also been introduced in the revised manuscript.

[9] Tan X, Zhi Q, Ma J, Chen Y, Li Y. Simulation of temperature and weld growth mechanism in ultrasonic welding of carbon fiber reinforced polyamide 66 composite: Employing the high frequency real-time horn vibration[J]. Journal of Materials Research Technology, 2023, 27: 5559-5571.

[10] Zhi Q, Li Y, Shu P, Tan X, Tan C, Liu Z. Double-pulse ultrasonic welding of carbon-fiber-reinforced polyamide 66 composite[J]. Polymers, 2022, 14: 714

[11] Tan X R, Zhi Q, Li Y B, Chen Y H, Chen Y Q, Shi X H, Wu H Y. Effect of MoS2 on microstructure and mechanical property of ultrasonically welded carbon fiber/polyamide 66[J]. Journal of Materials Research Technology, 2024, 29: 2857-2867.

  1. Has the temperature variation during the welding process not been evaluated? Why? wouldn't it be important for the discussion of the results and conclusions?

Response 8:The temperature evolution is indeed very important in ultrasonic welding of ABS. We measured the transient temperature by embedding the thermocouples in the upper workpieces in previous other studies, and the results indicated that the measured temperature was close to the simulated ones at early phase of ultrasonic welding and showed certain differences in the late phase [9, 11]. In this study, we try to measure the transient temperature during welding, but the results are not reliable. The heating rate in ultrasonic welding of ABS enhances significantly comparing to that of polyamide 66. Based on previous studies [9, 11], the effective information was mainly provided by the early stage of the process where the workpiece still maintained their stiffness or part of the harness. For amorphous ABS, stiffness of the composite decreases dramatically along with its quick softening, the measured temperature is not reliable due to the reaction of the thermocouple, friction, collapse of the workpieces, etc. Therefore, the effective information provided by measuring the temperature is limited and is not presented in this study. Besides, we are still trying to find other available techniques to collect the temperature variation during the welding process.

  1. Figure 9 - Are these conclusions only obtained based on numerical simulation? How can it be compared with the experimental ones? Are only the results of the numerical simulation enough to draw conclusions?

Response 9: We are sorry that the conclusion is poorly arranged. The conclusions have been rewritten and it obtained based on experimental results and simulations. The numerical model is validated by comparing the size of the weld area and cross section morphologies of the joint as seen in Figs. 8, 11, 12, 13 and 14.

  1. Figure 15 is related to the FTIR result, but during the article no reference is made to FTIR.

Response 10:The reviewer’s comments are well-taken. Related references have been cited as References 37 and 38 in the revised manuscript.

  1. The results must be presented and discussed. Were the results expected or not? Why? Why not?

Response 11:The reviewer’s comments are well-taken. These results are expected and described in the revised manuscript.

  1. Poor discussion with open literature. The conclusions may be questionable.

Response 12: The reviewer’s comments are well-taken. Discussions section has been improved and compared with more open literature.

Round 2

Reviewer 4 Report

Comments and Suggestions for Authors

The authors took my suggestions into consideration.